



# Uncertainty in Offshore Wind Power Forecasts: A Regional Climate Modeling Approach for the North Sea

Alberto Elizalde[1], Naveed Akhtar[1], Beate Geyer[1], and Corinna Schrum[1,2]

[1]Institute of Coastal Systems - Analysis and Modeling, Helmholtz-Zentrum Hereon, Geesthacht, Germany
[2]Center for Earth System Research and Sustainability, Institute of Oceanography, University of Hamburg, Germany

**Correspondence:** Alberto Elizalde (alberto.elizalde@hereon.de)

**Abstract.**

With the transition towards green energies gaining momentum, the expansion of wind farm areas and associated technologies is growing faster. The North Seas Energy Cooperation group has set an ambitious target to increase the offshore wind-generated power capacity from 26 GW in 2022 to 300 GW by 2050 in the geographical areas of the North Seas. With this goal, an

extensive offshore infrastructure is planned to be deployed in the region. Studies have been carried out to assess the power production of such future development. However, the uncertainty of such assessments has not been fully addressed. Wake effects have been identified as the primary source of power losses. They are often studied within individual wind farms or small clusters, but the dynamics of large wind farm clusters at a regional scale are only beginning to be explored. In this study, we address uncertainties of power output derived from projected wind farm areas at the North Sea in scenarios that encompass

different turbine setups and atmospheric conditions. To achieve this, we used COSMO6.0-CLM, the newest version of the regional climate model COSMO-CLM, and further improved the existing wind farm module to extend the model's capability to design more flexible and realistic scenarios. This allows us to quantify impacts from different factors that contribute to power output uncertainties. Our results show that wake dynamics resulting from different turbine density distributions can account for up to 5% of the variability of the generated power, while wind regimes at different hub heights contribute an additional 2%.

Approximately 6% of the variability is attributed to discrepancies in atmospheric circulation states inherent to the reanalysis datasets used to force the simulations. The total uncertainty in power output accounts for 13%. In a scenario with an installed capacity of 150 GW the total power output would range from 58 to 74 GW, corresponding to an uncertainty of 20 GW. Since economic and environmental studies rely on such scenarios, it is crucial to consider these uncertainties.

## 1 Introduction

The ambitious goal of the European Union to be climate-neutral by 2050 has triggered an increasing demand for the production of renewable energy. Many European countries have expanded their plans to use wind energy as part of the solution (NSEC, 2021; Bundesregierung, 2022). According to the renewed political declaration from the North Seas Energy Cooperation group (NSEC), the total offshore installed capacity in Europe is planned to further increase from approx. 26 GW in 2022 to at least 60 GW by 2030 and 300 GW by 2050 (NSEC, 2021). As part of the strategy to achieve this goal, the development of large





wind farm clusters across the North Sea region is being fostered. However, such a large-scale deployment will induce mutual influence between wind farms, which may negatively impact their performance, leading to a greater reduction in electricity production than initially anticipated. Uncoordinated development could also lead to significant losses of expected electricity generation. Developments in multiple countries could influence each other, requiring international treaties on transboundary resources, similar to freshwater, oil, and fisheries (Lundquist et al., 2018). Moreover, previous studies have shown that the

effects of the wakes produced from wind farm areas can cause local and regional changes in the marine environment. For example, impacts on the atmospheric circulation (Hasager et al., 2015; Akhtar et al., 2021), the ocean circulation (Ludewig, 2013; Christiansen et al., 2023) and the ecosystems (Daewel et al., 2022; Gușatu et al., 2021). Therefore, a comprehensive understanding of wake interactions at regional scales, particularly in large wind farm clusters, is essential for assessing their impacts and associated uncertainties.

Numerous studies based on observations and numerical simulations have been conducted to examine the implications of wake effects, especially within the energy sector, where the generated power losses play a significant role in economic planning. The generated power can be easily compared with the installed capacity to derive the wind farm efficiency (also called capacity factor). Given that the sector is still under development, those studies are often focused on a single turbine or wind farm only. Reports indicate values of capacity factor from single onshore wind farms of 26% in the UK (Waters, 2023), 55% in Poland

(Olczak and Surma, 2023), and from 21% to 55% (with an approximated average value of 35%) in the US (Wiser et al., 2022). Whereas for offshore wind farms in Europe and US, the capacity factor is estimated from 23% to 52% (with an average of 35%) (Cai and Bréon, 2021; WEO, 2019; Menezes, 2019; Stehly et al., 2018; Waters, 2023; Cassa, 2024; Smith, 2024). However, studies that address power losses in clusters are rather limited; Nygaard (2014) and Nygaard and Hansen (2016) found that, using observational data from a wind farm in Denmark, its efficiency was reduced up 10 to 16% in the direction range from

where a neighboring wind farm was newly constructed. Lundquist et al. (2018) estimated a power loss of downwind wind farms by 5% from observation data for a specific month (January 2013). A more comprehensive approach is presented in Akhtar et al. (2021), where a 10 year transient climate simulation incorporates single-size turbines (5 MW) distributed across operational and planned wind farm sites in the North Sea using a numerical model with the Fitch wind farm parametrization (Fitch et al., 2012). They estimate an average reduction of the capacity factor of 20% for the North Sea wind farms. In more

recent studies, Akhtar et al. (2024) and Borgers et al. (2024) compare the increment on the capacity factors by upgrading the rated power of the simulated turbines from 5 MW to 15 MW, Akhtar et al. (2024) report an increment of 2 to 3%, where as Borgers et al. (2024) estimates the increment by about 9%. The variability can be attributed to the distinct nature of each study; however, a fair comparison of the results is challenging due to differences in locations, wind farm configurations, wake calculation methods, atmospheric conditions, turbine specifications, and other factors, but at the same time, this highlights the

significance of the uncertainty in such assessments.

To investigate and quantify the variability and uncertainty of wake effects, we conducted simulations following a similar approach to Akhtar et al. (2021). The numerical model was further refined to account for different turbine specifications and to integrate wind farm metadata (Elizalde, 2023). These enhancements enable the representation of more realistic wind farm configurations, incorporating a nonhomogeneous distribution of turbine characteristics and time based activation of wind





farms under otherwise identical conditions. To validate the model, a spatial and temporal variability analysis of wind speed was performed. Additionally, to capture uncertainties arising from atmospheric conditions, each scenario was simulated using two different reanalysis datasets (ERA-Interim and ERA5). All simulations assumed a total installed capacity of 150 GW for the North Sea region, consistent with the mid-range targets outlined in the NSEC plans.

The structure of the paper is as follows: Section 2 describes the data sources used for turbine specifications, wind farm metadata, and observed wind speeds for model validation. Section 3 presents enhancements to the wind farm scheme within the COSMO-CLM climate model (Rockel et al., 2008) and details the simulated scenarios. Model validation, as well as the analysis of wake effects and power losses, are discussed in Sect. 4. Finally, conclusions are provided in Sect. 5.

## 2 Data

### 2.1 Turbines specifications

As part of the new approach in our wind farm simulations, we enable within the atmospheric model COSMO6.0-clm the distribution of different turbine specifications (power capacity, hub height, and rotor diameter) over the simulated domain. Turbine information was obtained from the National Renewable Energy Laboratory Turbine Archive (NREL, 2020) and Wind Turbine Model database (WTM, 2020) and listed in Table 1. The databases provide turbine dimensions, tabular curve data for wind speed-dependent idealized power output and thrust, and power and thrust coefficients for onshore and offshore turbines. The coefficients correspond to those used in the wind farm parametrization to calculate momentum sink, turbulent kinetic energy, and power output (Sec. 3). However, NREL (2020) reports that power curves calculated from power coefficients deviate from idealized power output due to "a number of reasons" without further details. This discrepancy was corrected by scaling the provided power coefficients to fit the expected values. The scale factor (the ratio between the idealized power and uncorrected power above-rated wind speeds) for each turbine is shown in Table 1, and the respective power curve is depicted in Fig. 1. The unevenness of the calculated power curves at above-rated wind speeds is the result of larger gaps in the provided uncorrected tabulated coefficients. Modern turbines have power curves that reflect control strategies around the cut-off wind speed, allowing them to operate beyond such a threshold. These control strategies are not considered in the NREL or WTM data, nor included in the parametrization developed here. In our case, the turbine shutting is simulated by setting to zero the power coefficient above the cut-off speeds, and by using the last value before the cut-off for the thrust coefficient.

### 2.2 Wind farm metadata

The wind farm metadata was updated from the previous model version. The European Marine Observation and Data Network information (EMODnet, 2022) supported by the European Union's Integrated Maritime Policy has been incorporated in the COSMO6.0-clm model. This open-source dataset is continuously maintained on a monthly basis with new information as it becomes available. The data used in this study corresponds to the version available at the time of access in March 2022. The dataset provides information on the status of each wind farm area (in production, in construction, approved, planned, etc.) and




**Table 1.** Turbine specifications from NREL and WTM used for the atmospheric simulations with different technical scenarios. The scale factor denotes the correction applied to each turbine tabulated data from NREL and WTM databases to better align with their idealized rated capacity.

| Manufacturer | Rated power (MW) | Hub height (m) | Rotor diameter (m) | Scale factor | Source |
|---|---|---|---|---|---|
| Siemens SWT | 3.6 | 90 | 120 | 0.993 | WTM |
| NREL | 5 | 90 | 126 | 1.013 | NREL |
| LEANWIND | 8 | 110 | 164 | 1.011 | NREL |
| IEA | 10 | 120 | 198 | 0.94 | NREL |
| IEA | 15 | 150 | 240 | 0.938 | NREL |

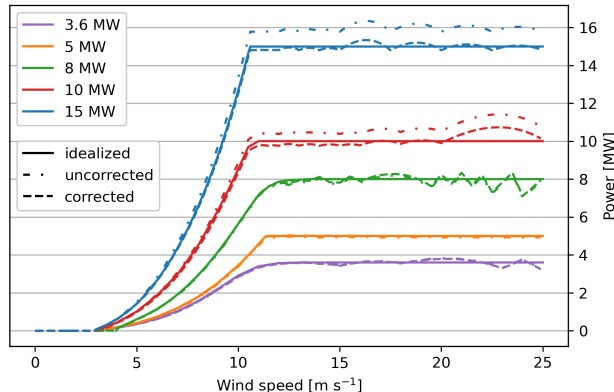

**Figure 1.** Power curves for different turbines. Idealized power (solid lines), uncorrected curves using turbine power coefficients (dashed-dotted lines), and corrected curves (dashed lines).

the geographical location of its boundaries. Detailed specifications are not available for all wind farms, especially not for those in planning status. In such cases, the turbine specifications for the highest rated capacity (15 MW) were used. Figure 2 shows the model domain and the distribution of the wind farm areas with their corresponding rated power based on EMODnet data.

### 2.3 Wind speed observations

The Advanced Scatterometer (ASCAT) satellite-based data (Ricciardulli and Wentz, 2016) is used to evaluate the spatial variability of the simulated wind. It is a modified version of the daily EUMETSAT MetOp-ASCAT ocean surface wind vector product v02.1 released by Remote Sensing Systems in April 2016 (Ricciardulli and Wentz, 2016). Remote sensing techniques are used to compute near-surface wind vectors at 10 meters height above the ocean surface by measuring the sea-surface backscattering signal due to the roughness of the sea surface (Gelsthorpe et al., 2000). It provides daily data from ascending

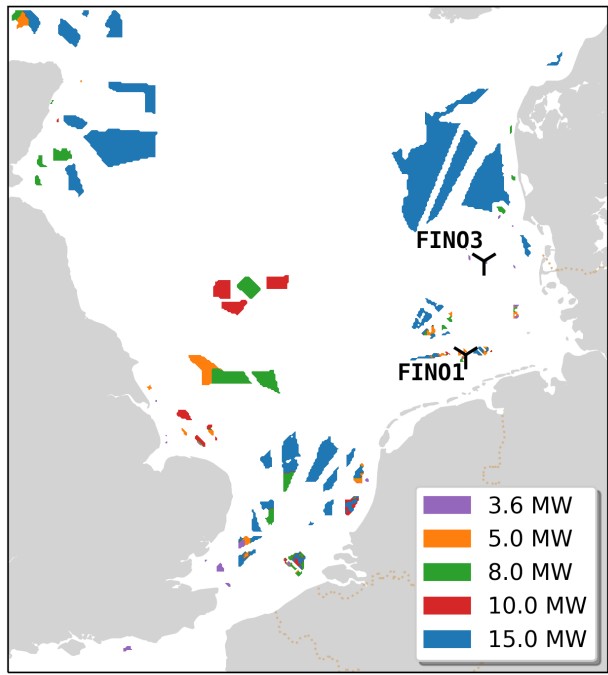

**Figure 2.** Spatial distribution of wind farm areas and their rated capacities according to EMODnet dataset (accessed on March 2022). The black markers show the location of observational platforms FINO1 and FINO3.

and descending passages. The dataset has a spatial resolution of 25 km and a temporal coverage period from 2007 to the present. For the model validation period from 2013 to 2018 we utilized data from the MetOp-A mission only.

For the validation of the temporal variability, hourly wind speed data is obtained from FINO1 and FINO3 research platforms (Westerhellweg et al., 2012; Leiding et al., 2016) in the North Sea provided by the Bundesamt für Seeschifffahrt und Hydrographie (BSH) (BSH, 2023). The data consist of mast corrected 10 min average values from sonic and cup anemometer

measurements installed at 8 to 9 heights from 21 to 107 meters on the masts. FINO1 is located at approximately 45 km north of Borkum at coordinates N 54° 00' 53.5" E 6° 35' 15.5" in the immediate vicinity of an operating wind farm cluster composed by Alpha Ventus, Borkum Riffgrund I and II, the westbound TranelWindpark Borkum and Mercury with other wind farms in the vicinity under construction. FINO3 is located 80 km west of Sylt with a potential influence of onshore wind turbines at the coast and nearby operating offshore wind farms of Butendiek, DanTysk and Sandbank.

Simulated wake effects were compared with airborne measurements from the German Research project WInd PArk Far Field (WIPAFF) (Emeis et al., 2016; Platis et al., 2020). The field campaign consisted of 41 flights in different wind farm locations at specific times and altitudes. The research aircraft recorded all components of wind speed, humidity, temperature, and pressure at a sampling frequency of 100 Hz. In this work, flight number 7, dated 10 September 2016 from 0800 UTC to 1100 UTC, was selected as it captured the main characteristics of the wake extension at hub level (90 meter) between the cluster formed by the





wind farms of Amrumbank West, Nordsee Ost, Windpark Meerwind SuedOst, and the Butendiek wind farm located at 50 km
at the north of the cluster (Baerfuss et al., 2019) making it a good case study for model results comparison (Sec. 4.3).

## 3   Methods

### 3.1   Atmospheric model

The atmospheric model COSMO-CLM (Rockel et al., 2008) is a non-hydrostatic limited-area model widely used in climate
research (e.g., Sørland et al., 2021). It describes a compressible flow in a moist atmosphere based on thermo-hydrodynamical
equations. The equations are solved numerically with a Runge-Kutta time-step scheme (Wicker and Skamarock, 2002) at meso-
$\beta$ and meso-$\gamma$ scales (2 to 200 km) (Steppeler et al., 2003) on a three-dimensional grid according to the Arakawa-C scheme
with a Lorenz vertical grid staggering (Arakawa and Lamb, 1977). The grid is defined on rotated geographical coordinates with
a terrain-following height coordinate (Doms et al., 2013). Initial and lateral boundary conditions are imposed by a driving host
model. At upper levels, a sponge layer with Rayleigh damping is used, whereas lateral boundary conditions use a 1-way nesting
by Davis-type formulation. The sea surface temperature is prescribed by the forcing data. Horizontal and vertical advection is
treated explicitly; tendencies are evaluated by taking advection velocities, kinetic energy and absolute vorticity at the centered
time level $n$ of the leapfrog scheme. The scheme for vertical turbulent transport at sub-grid scale is based on a second-order
closure at hierarchy level 2.0 (Mellor and Yamada, 1974). The standard physical parametrizations include the radiative transfer
scheme (Ritter and Geleyn, 1992), Tiedtke parametrization for convection (Tiedtke, 1989), and a turbulent kinetic energy-based
surface transfer and planetary boundary layer parametrization (Raschendorfer, 2001).

The wind farm parametrization developed in COSMO5.0-clm15 (Akhtar and Chatterjee, 2020) has been implemented in the
COSMO6.0-clm version (Elizalde, 2023) (see Sect. 3.2). COSMO6.0-clm is the final COSMO version and the result of the
unification of all developments in physical parametrizations of the numerical weather prediction and the climate mode. The
model structure of the physical parametrization was changed to the structure of the most recent numerical model ICON (Doms
et al., 2021). The updated schemes consist on the prognostic variables for the microphysics (water vapour, cloud water, ice,
rain. snow and graupel) (Doms and Förstner, 2004; Seifert and Beheng, 2001), radiation, sub-grid scale orography (Lott and
Miller, 1997), prognostic variable for the turbulence (Raschendorfer, 2001), surface schemes (TERRA (Schrodin and Heise,
2001), FLake (Mironov et al., 2010) and SeaIce (Mironov et al., 2012)) and convection (Tiedtke or shallow) (Tiedtke, 1989).
The simulations are dedicated to the North Sea region (Fig. 2). The model domain consists of 365 x 396 grid points at a
spatial resolution of 0.02° (approximately 2.2 km), with 20 grid points allocated to the lateral sponge zone and 61 vertical
levels extending up to 22 km. The rotated north pole is located at 180 W, 30 N. The temporal resolution of the model output
was configured to hourly intervals.



## 3.2 Wind farm parametrization

The Fitch et al. (2012) wind farm parametrization has been successfully used in previous COSMO-CLM model versions (Chatterjee et al., 2016; Akhtar and Chatterjee, 2020; Akhtar et al., 2023). In this work, we implemented it from COSMO5.0-clm15 (Akhtar and Chatterjee, 2020) to the most recent version COSMO6.0-clm (Elizalde, 2023). The implementation consisted of fitting the wind farm scheme into the updated physical parametrization of COSMO6.0-clm and, in addition to improvements for model performance, the calculation of the power output has been introduced following the same method as in Fitch et al. (2012). Momentum, TKE tendencies, and generated power of a turbine at the model layer $k$ are calculated as follows:

$$\frac{\partial \boldsymbol{u_k}}{\partial t} = \frac{1}{2} \frac{N_T A_k C_T U_k \boldsymbol{u_k}}{(z_{k+1} - z_k)} \tag{1}$$

$$\frac{\partial TKE_k}{\partial t} = \frac{1}{2} \frac{N_T A_k C_{TKE} U_k^3}{(z_{k+1} - z_k)} \tag{2}$$

$$P_k = \frac{1}{2} N_T A_k C_P U_k^3 \rho_k \Delta x \Delta y \tag{3}$$

where $TKE$ is the turbulent kinetic energy, $\boldsymbol{u}$ the wind the vector and $U$ the magnitude of wind speed on the horizontal plane, respectively, $P$ is the generated power, $N_T$ is turbine density calculated as the number of turbines per wind farm area, $A$ is the area of the rotor, $\rho$ is the air density, $\Delta x$ and $\Delta y$ are the grid size in the zonal and meridional directions, $z$ the height of the vertical level, $C_{TKE}$ is the turbulence coefficient calculated as $C_{TKE} = C_T - C_P$ with $C_T$ and $C_P$ being the thrust and power coefficients, respectively. We did not apply any correction to $C_{TKE}$ as suggested by Archer et al. (2020), $C_{TKE}$ remains as in the original calculation. After the wind farm tendencies are calculated, they are added to the general model tendencies. The power is accumulated according to the output frequency.

## 3.3 Wind farm scenarios

Different conceptual wind farm scenarios were designed as summarized in Table 2. Two scenarios were set to estimate the maximum difference of the wake effects based on the available turbine data; both scenarios featured homogeneous turbine types across all wind farm areas but with the smallest and the largest available turbines' rated power, respectively, i.e., 3.6 MW and 15 MW. A third simulation (Nonhomogeneous) consists of nonhomogeneous turbine types fitting the rated power for each wind farm as close as possible to the metadata information from the EMODnet dataset (Fig. 2). A fourth scenario (Chronological) setup follows the same configuration as the Nonhomogeneous scenario, but the wind farms were activated in the year each wind farm was commissioned, making this scenario more in line with real conditions in the past. For those wind farms that have no metadata on rated power, 15 MW turbines were assumed. Control scenarios with the wind farm parametrization switched off were used as reference simulations. They provide atmospheric data without accounting for wake effects. Control simulations





**Table 2.** Wind farm scenarios and the simulated periods based on the availability of the forcing datasets.

| Scenario name | Experimental setup | ERA-Interim | ERA5 |
|---|---|---|---|
| Control | No wind farm parametrization | 2008—2018 | 2008—2022 |
| Control-3.6 MW | Generated potential power | - | 2012—2018 |
| Control-15 MW | Generated potential power | - | 2012—2018 |
| 3.6 MW | All wind farm zones set with 3.6 MW turbines | 2008—2018 | 2012—2022 |
| 15 MW | All wind farm zones set with 15 MW turbines | 2008—2018 | 2012—2022 |
| Nonhomogeneous | Multi-turbine types | 2008—2018 | 2012—2022 |
| Chronological | Multi-turbine types with sequential activation | 2008—2018 | 2008—2022 |

(Control-3.6 MW and Control-15 MW) calculate the potential power generated for the 3.6 MW and 15 MW turbine scenarios, representing the power that could be produced in the absence of wake effects.

To investigate the uncertainty in the generated power due to atmospheric conditions, simulations were conducted using two different boundary forcing datasets: ERA-Interim (Dee et al., 2011) and ERA5 (Hersbach et al., 2020) reanalysis data, covering the periods from 2008 to 2022 depending on the driving datasets (Tab. 2). These datasets have spatial resolutions of approximately 0.7 degrees (approx. 79 km) and 0.285 degrees (about 37 km), respectively. To achieve the target resolution of 0.02 degrees (approx. 2 km), a two-step offline nesting approach was adopted. COSMO5.0-clm15 (without wind farm parametrization) was used as the initial step to downscale reanalysis data to a resolution of 0.11 degrees (approx. 12 km) (Geyer, 2014). Subsequently, this downscaled simulation served as driving conditions for the finer 0.02 degree resolution simulation conducted with COSMO6.0-clm, which includes the wind farm parametrization.

## 3.4 Atmospheric stability classification

There are many methods to classify the atmospheric stability based on the altitude of the process of interest. Here we follow the classification of the low atmosphere according to (Mohan, 1998), who established stability categories at turbulent boundary layer heights. This approach has been used in other studies related to wind energy research (Schneemann et al., 2020; Cantero et al., 2022). The calculated Bulk Richardson number ($Ri_b$) at hub height is used to estimate the stability based on wind speed and mean virtual potential temperature. We use a simplified version from Mohan (1998) where weakly unstable, weakly stable, and neutral categories from the original classification are all merged within the neutral category (Table 3). The motivation arises from the fact that, in our case, the results from these three categories are similar and do not provide additional information. Nevertheless, the reassignment of those categories simplified the analysis.

 

**Table 3.** Bulk Richardson number thresholds used for atmospheric stability classification.

| Stability class | Bulk Richardson number ($Ri_b$) |
|---|---|
| Very unstable | $Ri_b < -0.023$ |
| Unstable | $-0.023 \leq Ri_b < -0.011$ |
| Neutral | $-0.011 \leq Ri_b < 0.042$ |
| Stable | $0.042 \leq Ri_b < 0.084$ |
| Very stable | $0.084 \leq Ri_b$ |

## 4 Results

### 4.1 Wind speed validation

Wind speed observational data from FINO1 and FINO3 platforms were compared with simulated wind speeds for the common
period from 2012 to 2018. The time series at 90 meter height for the grid box of each FINO platform location was extracted
from the model grid and statistically compared (Table 4).

The FINO1 platform is located inside a cluster of 5 wind farms, with distances to the platform ranging from 1.4 to 15 km.
Within the comparison period, two wind farms were set into operation in 2015. The time series of FINO1 shows an increase
in turbulence intensity and a reduction of wind speed after each of the nearby wind farms became operational (Pettas et al.,
2021). In the entire period of FINO1 the yearly wind speed average decreased by around 1 m s$^{-1}$ from 2004 to 2019, mainly
attributed to the effect of the wind farm wakes (Ortensi et al., 2020).

Wind speed averages of the control experiments forced with ERA-Interim and ERA5, in comparison with FINO1 data,
overestimate the observed values by 0.22 and 0.64 m s$^{-1}$, respectively. A positive bias is expected, as the control simulations
do not include the wind farm parametrization and therefore do not account for wake effects. In contrast, for the scenario
simulations—except for the Chronological case—all wind farms are considered operational throughout the entire simulated
period. This includes all wind farms in the vicinity of the FINO1 cluster as well as those in the adjacent eastern cluster, which
amplifies the effects of wake effects and leads to an additional wind speed reduction, leading to a negative bias from -0.86
to -1.73 m s$^{-1}$ on those simulations. The Chronological simulation, which activates the wind farms near the FINO1 platform
sequentially according to their actual commissioning years, reduces the averaged wind speed bias to 0.08 m s$^{-1}$. This indicates
that this approach is capable to more accurately represent the wind conditions at FINO1. A more detailed comparison of the
temporal evolution is addressed in Sect. 4.2.

Similar to FINO1, FINO3 is located in the vicinity of three wind farms. Two of them were set into operation in 2015 and
a third one in 2017. The control simulation driven by ERA-Interim should show overestimations of the measured wind speed
at FINO3 location, given that FINO3 data is already under the influence of wind farm wakes. But only the ERA5-driven
simulations meet this expectation. A comparison with scenarios does not yield big differences. Same as in FINO1 results, the
results from the scenarios are influenced by the driving conditions used. Simulations driven with ERA5 not only consistently





**Table 4.** Statistical comparison of simulated wind speeds at 90 m with FINO data. The columns correspond to the average value ($\mu$), standard deviation ($\sigma$), 1st and 99th percentiles ($P_1$ and $P_{99}$), root mean square error ($RMSE$), Pearson correlation ($r$) and the p-value for the period of 2012 to 2018. The top and bottom sections correspond to the comparison with FINO1 and FINO3, respectively. All units are in m s$^{-1}$ except for $r$ and p-value, which are unitless.

| Dataset | $\mu$ | $\sigma$ | $P_1$ | $P_{99}$ | $RMSE$ | $r$ | $p - value$ |
|---|---|---|---|---|---|---|---|
| FINO1 | 9.24 | 4.58 | 1.44 | 21.75 | - | - | - |
| Ctrl. (ERA-Int.) | 9.46 | 4.46 | 1.2 | 21.19 | 2.41 | 0.86 | 0.0 |
| Ctrl. (ERA5) | 9.88 | 4.63 | 1.29 | 22.2 | 3.54 | 0.71 | 0.0 |
| 3.6 MW (ERA-Int.) | 7.51 | 4.06 | 1.19 | 20.53 | 3.02 | 0.84 | 0.0 |
| 3.6 MW (ERA5) | 7.94 | 4.32 | 1.33 | 21.49 | 3.66 | 0.71 | 0.0 |
| NH (ERA-Int.) | 7.68 | 4.39 | 0.94 | 20.83 | 2.91 | 0.85 | 0.0 |
| NH (ERA5) | 8.15 | 4.64 | 0.95 | 21.89 | 3.69 | 0.71 | 0.0 |
| 15 MW (ERA-Int.) | 7.93 | 4.39 | 1.06 | 21.36 | 2.74 | 0.86 | 0.0 |
| 15 MW (ERA5) | 8.38 | 4.63 | 1.14 | 22.24 | 3.56 | 0.72 | 0.0 |
| Chr NH (ERA5) | 9.32 | 4.71 | 1.14 | 22.30 | 3.51 | 0.71 | 0.0 |
| FINO3 | 9.68 | 4.61 | 1.35 | 21.93 | - | - | - |
| Ctrl. (ERA-Int.) | 9.68 | 4.53 | 1.23 | 21.54 | 2.50 | 0.85 | 0.0 |
| Ctrl. (ERA5) | 10.15 | 4.68 | 1.34 | 22.64 | 3.70 | 0.69 | 0.0 |
| 3.6 MW (ERA-Int.) | 9.42 | 4.53 | 1.24 | 21.54 | 2.50 | 0.85 | 0.0 |
| 3.6 MW (ERA5) | 9.91 | 4.7 | 1.41 | 22.55 | 3.64 | 0.70 | 0.0 |
| NH (ERA-Int.) | 9.44 | 4.52 | 1.28 | 21.58 | 2.49 | 0.85 | 0.0 |
| NH (ERA5) | 9.92 | 4.69 | 1.42 | 22.64 | 3.63 | 0.70 | 0.0 |
| 15 MW (ERA-Int.) | 9.51 | 4.54 | 1.23 | 21.65 | 2.49 | 0.85 | 0.0 |
| 15 MW (ERA5) | 9.99 | 4.7 | 1.4 | 22.68 | 3.65 | 0.69 | 0.0 |

produce higher wind speeds than those driven with ERA-Interim (as shown by the average and 1st and 99th percentiles values in Table 4), but also show higher variability as described by the standard deviation values. The $RMSE$ and Pearson correlation indicate that those higher speeds represent a larger deviation from measurements for both FINO platforms, suggesting that, in

our case, the ERA-Interim forcing is more beneficial to represent wind speed at FINO stations.

The spatial variability of the simulated wind speed field is validated against ASCAT satellite-based data. We included not only both of the control simulations but also the wind speed field from the reanalysis datasets and the in-between COSMO5.0 simulations (with $0.11°$ resolution) that were used to drive the high-resolution COSMO6.0 simulations.

The yearly cycle of the 10 meter wind speed over the North Sea is shown in Fig. 3. ERA5 and ERA-Interim climatologies

underestimate the 10 meter wind speed in general, with the largest bias during the winter months. COSMO simulations tend to

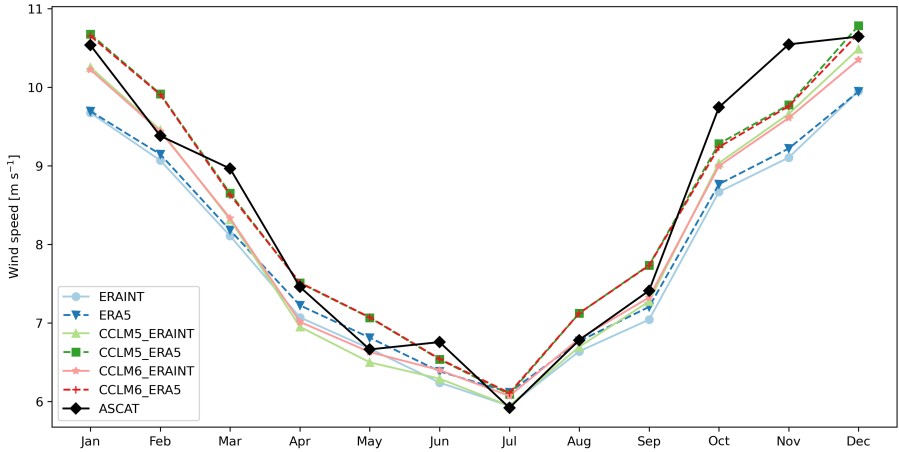

**Figure 3.** Yearly cycle for wind speed at 10 m height for the reanalysis data ERA-Interim and ERA5 (blue, solid and dashed), the satellite data ASCAT (black), and the control simulations done with the COSMO model versions 5 (green) and 6 (red), averaged over the North Sea for the period from 2013 to 2018. Near-coast areas are excluded due to the data coverage of ASCAT data.

correct this bias in late winter months. Similar to the findings of the FINO analysis, ERA5 shows systematically higher wind speed values than ERA-Interim, which are inherited by the COSMO simulations. The differences between the model versions of COSMO-CLM 5.0 and 6.0 are smaller than the differences derived from using different forcings, which indicates that the double nesting approach and the change of model version do not accumulate any errors due to model numerics in the results.

The spatial distribution of the $RMSE$ of the 10 meter wind speed based on daily values (see Supplemental Material Fig. S1) shows a larger error of around 1 m s$^{-1}$ in the eastern side of the North Sea for our control simulations. Larger biases in the coastal areas in the reanalysis datasets are due to their low resolution, which mixes wind-related effects from land and ocean areas. Overall, the $RMSE$ from ERA-Interim is smaller than in ERA5.



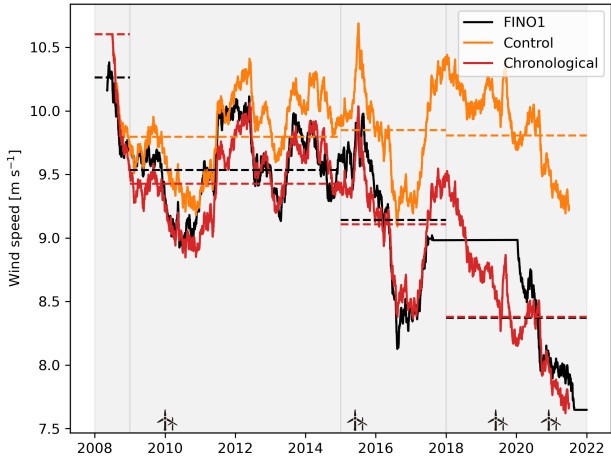

**Figure 4.** One year running mean time series for the mast corrected observed wind speed at 91 meters from the FINO1 platform (in black), simulated wind speed at 90 meters for the Control simulation forced with ERA5 (in orange), and for the Chronological simulation (in red). Wind turbine symbols denote the commissioning year of wind farms near the FINO1 platform. Dashed lines indicate the averaged values (Table 5) for the phases defined by the vertical lines. Observational data is unavailable for the period 2018—2020.

## 4.2 Reconstruction of wind speed at FINO1 platform

As new wind farms have been commissioned around the FINO1 platform over the years, wind speed measurements from the platform's onboard sensors provide an opportunity to quantify the impact of these developments on wind speed deficits and power generation within wind farm clusters. We compared observed wind speed with the Chronological simulation, in which wind farms were activated according to their commissioning year, with their specifications (area, turbine density, turbine height, rotor diameter, and rated power) adapted as closely as possible to reality based on EMODnet metadata. Figure 4 shows the

wind speeds within the period 2008—2022 and the commissioning years of the wind farms in the vicinity of FINO1 (marked by small turbine symbols). The Control simulation represents the natural variability of wind speed without wake effects (orange curve), showing no clear trend during the analyzed period. However, the negative trend in observed wind speed can only be attributed to the presence of the new wind farms (black curve). This negative trend is well captured in the Chronological simulation (red curve). The entire period was divided into four phases, with time boundaries close to the commissioning years of the wind farms. Table 5 summarizes the average wind speed in each phase and the corresponding power generated by each

wind farm. Wake effects contribute to a wind speed reduction of up to 16% at the FINO1 platform by the end of the period, leading to an estimated 18% decrease in power production at the Alpha Ventus wind farm.



**Table 5.** Averaged simulated power output (MW) for several wind farms and wind speed (m s$^{-1}$) at FINO1. The phases were defined by the commissioning years of the wind farms.

| Power Output (MW) | | | | | |
|---|---|---|---|---|---|
| Wind farm | Commissioning year | phase one 2008—2008 | phase two 2009—2014 | phase three 2015—2017 | phase four 2018—2022 |
| Alpha Ventus | 2010 | - | 32.8 | 30.1 | 26.1 |
| Triane Borkum 1 | 2015 | - | - | 69.3 | 61.7 |
| Borkum Riffgrun 1 | 2015 | - | - | 159.1 | 140.3 |
| Merkur | 2019 | - | - | - | 156.9 |
| Borkum Riffgrun 2 | 2019 | - | - | - | 196.2 |
| Triane Borkum 2 | 2020 | - | - | - | 61.4 |

| Wind speed (m s$^{-1}$) | | | | |
|---|---|---|---|---|
| Dataset | phase one | phase two | phase three | phase four |
| FINO1 | 10.1 | 9.5 | 9.1 | 8.5 |
| Control (ERA5) | 10.4 | 9.8 | 9.8 | 9.9 |
| Chronological (ERA5) | 10.3 | 9.4 | 9.1 | 8.4 |

## 4.3 Case study of a wind farm cluster

### 4.3.1 Wake extension

The simulated wake effects are validated by using airborne measurements from the WIPAFF project. The observed wind speed data between 8:20 UTC and 09:35 UTC on the 10 September 2016 is compared with the simulated wind speed from the 3.6 MW scenario at 09:00 UTC on the same day. A low-pressure system centered over the Faroe Islands on that day favored southwesterly winds at the German Bight and presented stable atmospheric conditions (Siedersleben et al., 2018) wind an average wind speed of 7.86 m s$^{-1}$. The wake induced by the Amrumbank West, Nordsee Ost, and Windpark Meerwind 255 SuedOst cluster was advected northwards with a prevailing wind direction of approximatelly 188°, towards the Butendiek wind farm. The airborne data has a 10 milliseconds temporal resolution. To compare with the simulation, the observed time series was smoothed by performing a running mean with a 3184 point window equivalent to a 2.2 km distance that matches the model spatial resolution. The model simulation is able to capture the wake effects as shown by wind speed anomalies in Fig. 5. The downstream wind deficit closest to the cluster is about 1 m s$^{-1}$ in both (observed and simulated data) and decreases 260 gradually northwards. The wind deficit extends up to 50 km, reaching the surrounding area of the Butendiek wind farm.





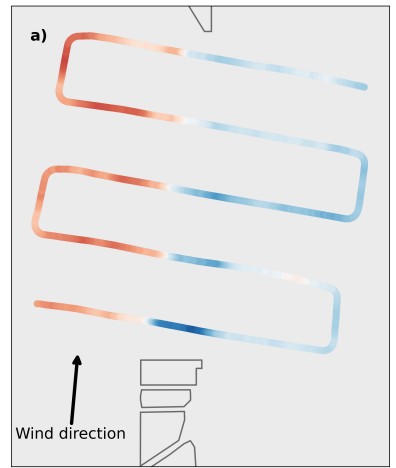 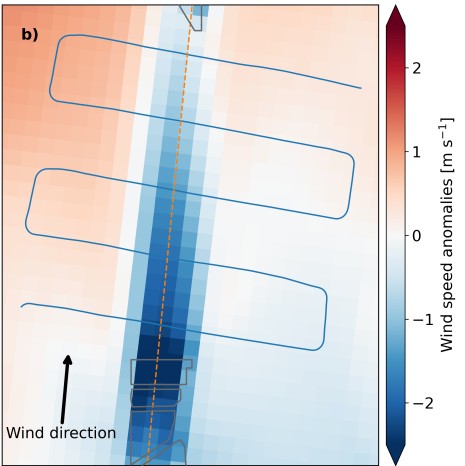

**Figure 5.** Anomalies for wind speed from airborne observations (left) and simulated data from the 3.6 MW scenario (right) in the downstream area of the wind farm cluster Amrumbank West, Meerwind Süd/Ost, and Nordsee Ost and the Butendiek wind farm in the north (in solid gray) on the 10 September 2016 at 09:00 UTC. The red dashed line marks the cross-section depicted in Fig. 6.

.

The overall wind speed average is underestimated by approximately 1 m s$^{-1}$, similar to results performed with the model version COSMO5.0-clm15 (Akhtar and Chatterjee, 2020). This bias is not related to the wake, but the general bias explained in the validation section (Sec. 4.1). The east-to-west gradient outside of the wake region is not well captured by the model, particularly on the eastern side. This bias seems to be not specific to COSMO6.0 only. A similar pattern has been reported in
simulations using COSMO5.0-clm15 (Akhtar and Chatterjee, 2020) and WRF models (Siedersleben et al., 2018).

Fig. 6 shows the wake influence on the vertical structure of the airflow across the section depicted in Fig. 5 (red dashed line). Wind farm areas are denoted by the space between the gray dashed lines, where the AW-MS-NO cluster is depicted on the left of each image, and Butendiek in the north is on the right side. The comparison refers to the difference between each scenario (3.6 MW, NH, and 15 MW) and the control simulation. The largest wind deficits are located at the lee side of the
cluster at altitudes between the hub height and the top of the rotor with values up to -2.9 m s$^{-1}$ in the 3.6 MW scenario and up to -2.8 m s$^{-1}$ for the 15 MW one, which represents an approximatelly 45% reduction with respect to the control simulation. The difference between the scenarios is due to the presence of the largest number of turbines in the 3.6 MW scenario to achieve the same installed capacity as in the 15 MW scenario. The influence of the wake is not limited to the top of the rotor, but reaches higher altitudes. For the smaller turbines of 3.6 MW, the deficits extend up to 100 meters above the rotor to an altitude
of 250 meters. In the 15 MW scenario, the influence of the wake is extended only 50 metes higher, to a height of approx. 300 meters. The influence on higher altitudes of similar magnitudes has also been reported in previous studies (Akhtar et al., 2022; Siedersleben et al., 2020). The wake recovers as the distance from the cluster increases, starting from the upper layers where the wake influence is smaller. At the southern border of the Butendiek wind farm, the simulated wind deficit is limited to a height of 200 meters in all the scenarios, and it accounts for about 10% of wind speed reduction.

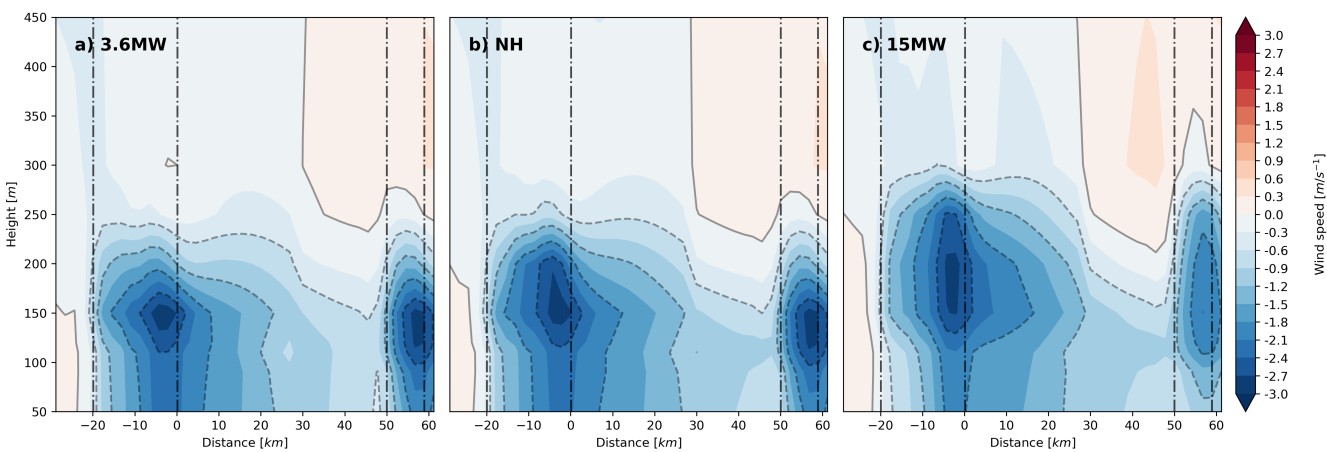

**Figure 6.** Wind speed differences due to wind farms (m s$^{-1}$) along the cross-section through the cluster and neighbor wind farm for three scenarios (3.6 MW, nonhomogeneous rated capacity (NH), and 15 MW) compared to the reference simulation without wind farm parametrization for Sep. 10, 2016. The airflow is directed from south to north (left to right in the image). The boundaries of the wind farm areas are indicated by vertical dash-dotted lines. Negative values are accentuated using dashed contour lines, while the zero contour is shown as a solid line.





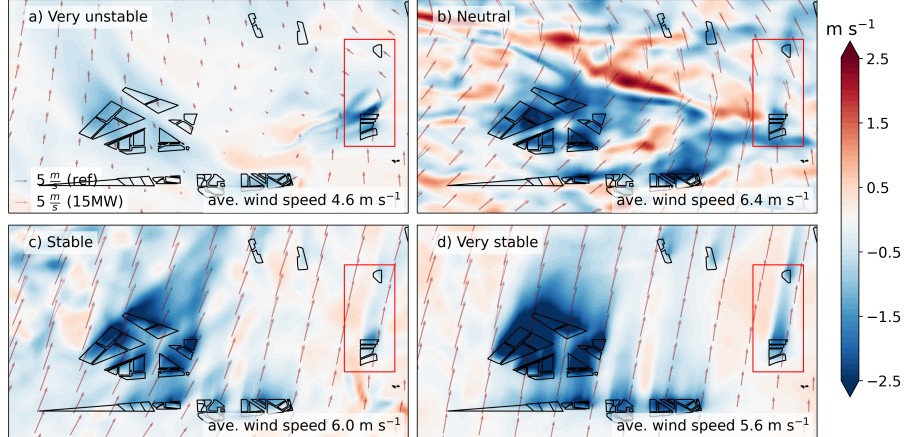

**Figure 7.** Composites of wind speed deficits (m s$^{-1}$) at 150 meters height for southwesterly winds (183 - 193°) classified by their atmospheric stability at the German Bight area for September, 2016. The wind direction is depicted by vectors in black for the reference simulation and in red for the 15 MW simulation. The wind farm cluster case study area (Fig. 5) is highlighted in the red box. The legends provide wind speed averages over the Amrumbank West wind farm.

### 4.3.2 Influence of atmospheric conditions on wake extension

It is well known that wake extension depends on atmospheric stability: unstable conditions enhance mixing and turbulence intensity, which increases the momentum transfer that diminishes the wind speed deficits (Platis et al., 2022). Whereas during stable conditions, laminar flow advects unperturbed wind speed deficits, which increases the wake extension. Figure 7 shows composites of wind speed deficits based on different atmospheric conditions for the 15 MW scenario driven with ERA5 for September 2016. The constituents were filtered based on a wind direction criterion at the Amrumbank West wind farm (the northernmost wind farm in the cluster, located within the study case indicated by the red box). The wind direction should have a southerly component in the interval of $\pm 5°$ centered at $\delta = 188°$, i.e., in the range between 183° and 193°. The constituents were then classified by their respective atmospheric stability condition at 120 meter height (the closest height to 150 meters hub height at which the model internally calculates the Bulk Richardson number) according to Table 3. The figure shows the differences between the wind speed of the 15 MW scenario and the control run. The vectors indicate the mean wind fields of both simulations (every 10 grid boxes). The averaged wind speed at the Amrumbank West wind farm in all stability cases varies between 4.6 m s$^{-1}$ in the very unstable case and 6.4 m s$^{-1}$ in the neutral case; this implies that the wake extension is not necessarily coupled with the wind speed.

The figure also shows a larger group of clusters situated in the west of the case study, where most of the wind farm sites are still planned (at the time of writing). A significant large wake influence among those clusters is not limited only to stable conditions but also under neutral conditions as a consequence of their respective proximity. The averaged wind speed deficit





in the large cluster accounts for -3.1 m s$^{-1}$ during neutral conditions and up to -4.0 m s$^{-1}$ under very stable conditions, which evidences that a large impact on power production for such clusters is expected.



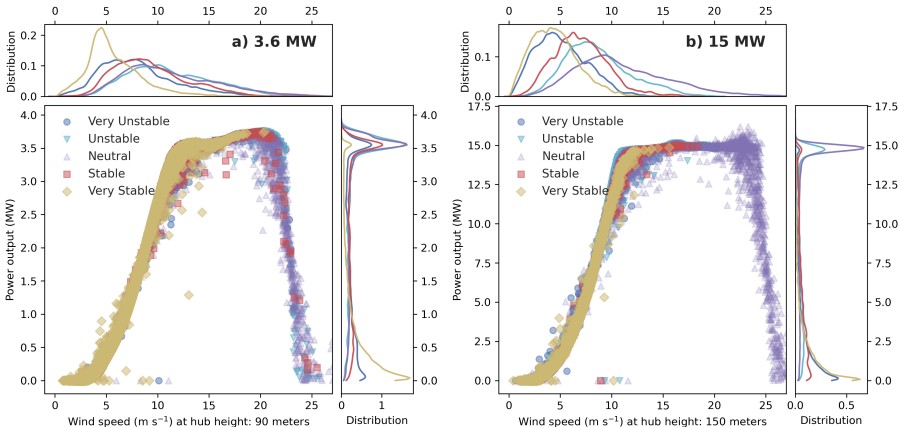

**Figure 8.** Power curves of hourly averaged data from 3.6 MW (a) and 15 MW (b) turbine types in the period from 2008 to 2018. The power output is classified by atmospheric stability conditions (in colors, plotted from very unstable to very stable, which causes overlaying symbols). Statistical distributions for the power output and wind speed are additionally depicted in the respective figure axes.

## 4.4 Turbine Power

Power production depends not only on wind speed but also on the atmospheric conditions and the influence of wakes from neighboring wind farms. To quantify the effects of atmospheric stability on the power output, the generated power has been classified according to atmospheric stability. Figure 8 shows classified simulated power outputs for the 3.6 and 15 MW turbine scenarios at the Amrumbank West wind farm, along with statistical distributions for power and wind speed derived from hourly time series over 11 year simulation period from 2008 to 2018 (in contrast to the composite analysis of a single month presented

in the previous section).

Wind speed values are more likely to occur within the turbine's partial load range (the wind speed interval between the cut-in, approx. 4 m s$^{-1}$, and the rated power, approx. 11 m s$^{-1}$) for all stability cases, as shown in the wind distribution figures. Nevertheless, power distribution figures indicate that the optimal turbine power output (i.e., rated power) occurs more often under neutral, unstable and very unstable conditions. Particular relevant are the stable and very stable cases, under which

relatively low wind speeds in the range from 4 to 11 m s$^{-1}$ may be further reduced if the turbine is under a wake influence, thereby diminishing the power production.

Simulated power values exceeding the turbine's rated power result from deviations in the provided tabulated data from the idealized power curve (see Fig. 1). The model does not account for turbine optimization above the cut-off wind speed (25 m s$^{-1}$); in such cases, the instantaneous power output is set to zero. However, the nonzero values above the cut-off wind speed

in the figure arise from the calculation of the hourly average, where not all wind speed values within an hour necessarily exceed the cut-off threshold.



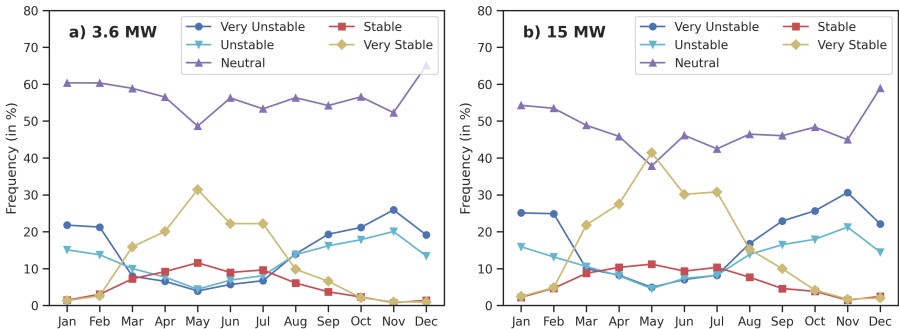

**Figure 9.** Yearly cycle of occurrences for the different atmospheric conditions in the Amrumbank West wind farm at a) 90 meter height for the 3.6 MW scenario and b) 150 meter height for the 15 MW scenario in the period from 2008 to 2018.





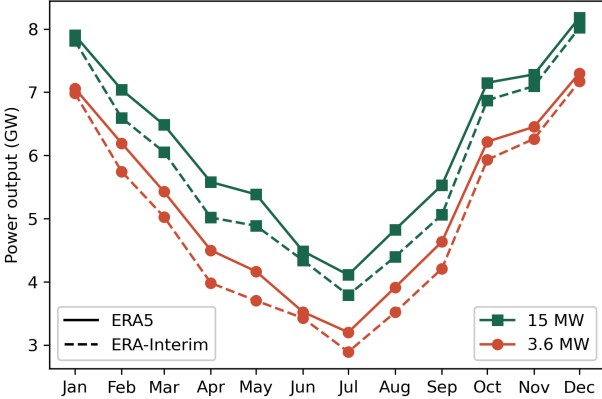

**Figure 10.** Yearly cycle for simulated power output (GW) from all wind farms in the North Sea for the period 2012—2018 for the two scenarios (3.6 and 15 MW in green and orange, respectively), driven by ERA5 and ERA-Interim forcing datasets, represented by solid and dashed lines, respectively.

### 4.5 Annual conditions for power production

Atmospheric stability cases do not occur with the same frequency throughout the year. Histograms for atmospheric conditions at 90 and 150 meters at wind farm sites in the North Sea from 2008 to 2018 show that neutral conditions prevail throughout

the year (Fig. 9). Stable and very stable atmospheric conditions are more prevalent during summer and the transitional seasons, with very stable conditions occurring more frequently. Turbines with a hub height of 150 m are exposed to a different wind regime, with more stable and very stable conditions than turbines with 90 m hub height. As depicted in Fig. 3, the averaged wind speed is at its lowest during summer months, therefore, the combination of low wind speed and laminar flow is more likely to induce larger power losses as a consequence of the wake effects at this time of the year.

Fig. 10 shows the annual distribution of power output for four simulations, the two scenarios (3.6 MW and 15 MW) driven with ERA5 and ERA-Interim datasets each. The yearly cycle of power output closely follows the seasonal pattern of wind speed (see Fig. 3). Nevertheless, two main features can be highlighted. Firstly, simulations driven with ERA5 boundary conditions produce higher power output than those driven with ERA-Interim. This is a consequence of the higher wind speeds in the ERA5 dataset as shown in the validation analysis (Sec. 4.1). The yearly mean wind speed is about 0.11 m s$^{-1}$ higher in ERA5

than in ERA-Interim. This corresponds to an annual power difference of approximately 4 GW more in ERA5 (approx. 6% of the generated power). Secondly, by design, the wind farms in each scenario were set to have the same installed capacity in the simulation domain regardless of the turbine type. Nevertheless, Fig. 10 shows that the 15 MW scenario has a higher power output than the 3.6 MW one. The reason lies in the different turbine densities between scenarios. To achieve the same rated power, the turbine density has to be higher in the 3.6 MW scenario than in the 15 MW one. The larger turbine density induces

more wake effects that reduce the power output. Additionally, the 15 MW scenario includes taller turbines, which are exposed to a different wind regime with higher speeds that produces higher power output.





**Table 6.** Annual power (GW) for the North Sea (see Fig. 2) with and without wake effects for the 3.6 and 15 MW scenarios driven with the ERA5 dataset for the period from 2012 to 2022. Row-wise differences (in %) refer to turbine size impacts. Column-wise differences (in %) refer to wake effects impacts. All percentages are calculated with respect to the total installed capacity of approximately 150 GW. Negative percentages imply a power output reduction.

|  | 3.6 MW [GW] | 15 MW [GW] | diff. [%] |
|---|---|---|---|
| No wakes | 85.9 | 93.9 | 5.3 |
| Wakes | 62.6 | 73.7 | 7.4 |
| Wake impact diff. [%] | -15.5 | -13.5 | - |

To quantify individual impacts from the turbine densities and turbines' types, two additional simulations driven with ERA5 forcings for each wind farm scenario (3.6 MW and 15 MW) were carried out to calculate the potential power output when neglecting the wake effects (no momentum sink and no TKE source effects). Table 6 summarizes the yearly potential power output of the North Sea wind farms with wakes and no wakes effects. The wake effects have an impact of 16 to 14% reduction of available power in the 3.6 MW and 15 MW scenarios, respectively. Therefore, the wake effects induced by the larger turbine density lead to 2% stronger reduction of power output (difference between both scenarios), whereas the difference due to wind conditions by taller turbines is 5.3%. The combined effects result in a higher power output of 7.4% in the 15 MW scenario.

The load factor of a wind farm, also known as the capacity factor, is defined as the ratio of the generated power to the installed capacity. Table 7 shows the averaged load factor of all simulated wind farms in the North Sea with a total installed capacity of 150 GW for each of the scenarios driven by the ERA5 dataset. Wind conditions throughout the year limit the produced power to a fraction of 0.57 and 0.61 of the installed capacity for the 3.6 MW and 15 MW scenarios, respectively. By including the wake effects, in the 3.6 MW scenario, the load factor is reduced further to 0.42. This is consistent with reported load factors in the range of 0.23 to 0.52 with an average value of 0.35 for the US and North Sea regions (Cassa, 2024; Smith, 2024). The discrepancy between the value in the 3.6 MW simulation and the reported averaged values in existing literature may be partially ascribed to the control and management of power output within real wind farms. Turbine shutdowns encompass actions such as the regulation of power output for adaptation to real-time electricity demand, maintenance, and equipment upgrades, among others. These actions, which are difficult to assess due to the large unpredictability and the lack of public data, are not integrated into the model.

However, by examining the difference between the 3.6 MW and 15 MW scenarios, the turbine type upgrade leads to an increment of the averaged load factor from 0.42 to 0.49, respectively. This represents an increment in the power output by 7%. This result stands in the range of recently reported increments of 2 to 3% in Akhtar et al. (2024), and 8.7% in Borgers et al. (2024) between 5 MW and 15 MW scenarios using COSMO5.0-clm15 version.

Equivalent load factors, based on ERA-Interim as driving conditions, can be estimated under the assumption of a 6% reduction in power output compared to ERA5, as described earlier. Therefore, the calculated load factors result in 0.39 and 0.46 for the 3.6 MW and 15 MW scenarios, respectively.



**Table 7.** Averaged load factors (as a fraction) for the North Sea with a total installed capacity of 150 GW based on simulations driven with ERA5 boundary conditions for the period from 2012 to 2022.

|          | 3.6 MW [frac.] | 15 MW [frac.] |
| -------- | -------------- | ------------- |
| No wakes | 0.57           | 0.61          |
| Wakes    | 0.42           | 0.49          |



## 5    Conclusions

The wind farm parametrization of Fitch et al. (2012) was implemented in the regional climate model COSMO6.0-clm, including the calculation of generated power. Additionally, the model now incorporates metadata for the wind farms and various turbine types, enabling flexible design for different wind farm scenarios. This enhancement enables the use of different turbine types within the same simulation, facilitating the creation of realistic scenario designs.

Similar to the previous model version, COSMO5.0-clm15, the wind field is simulated realistically, as confirmed by comparisons with station and satellite data. This marks an improvement over the wind speed values from the reanalysis products ERA-Interim and ERA5, which tend to underestimate wind speeds due to their coarser resolution, particularly in coastal areas where larger errors in the reanalysis data have been observed. The vertical structure and extent of the wake produced by the wind farms are also well represented according to airborne data.

It is well known that atmospheric flow with stable conditions enhances the wake extension by advecting the wind deficits over longer distances, with a potential impact on neighbouring wind farms. Our atmospheric stability analysis shows that future planned wind farms close to large neighbouring clusters can still be subject to wake effects in neutral or unstable conditions, even when the wake recovery is expected to be faster under turbulent conditions.

From the power analysis assessment, we found that 57 to 61% of the 150 GW installed capacity is available for power generation on account of weather regime and wind availability, with a further reduction of 15.5 to 13.4% due to wake effects in the 3.6 MW and 15 MW scenarios, respectively. Hence, in the quantification of power output uncertainties, we identified that around 5% is attributable to turbine types, due to difference in their heights and its exposure to different wind regimes. Approximately 2% is derived from the wake intensities due to the turbine density distribution. Finally, approximately 6% of the uncertainty is attributed to variations in the atmospheric state inherent in the reanalysis driving conditions. The total uncertainty of the power output accounts for approximately 13%, which, in a wind farm scenario with an installed capacity of 150 GW, results in a power output ranging from 58 to 74 GW, corresponding to an uncertainty of 20 GW.

The results presented here emphasize the significance of power uncertainties, as generated power values are commonly used to assess the economic and environmental impacts of wind farms. Additionally, power uncertainties implicitly reflect uncertainties within the wind field, which plays a crucial role in other physical processes, such as air-sea interactions. These interactions affect the seawater column, which, in turn, impacts local marine ecosystems — an area that still requires detailed study.

*Code availability.*   The new implementation of the wind farm parametrization of Fitch et al. (2012) on COSMO6.0-clm is publicly available as a patch to the model on Zenodo repository (Elizalde, 2023).

*Data availability.*   The wind farm scenario simulations are publicly available on the World Data Center for Climate repository hosted at the Deutsche Klimarechenzentrum (DKRZ) (Elizalde et al., 2024). The wind farm metadata from EMODNET was obtained from the public



European Commission website (EMODnet, 2022). Turbine coefficient data were obtained from the National Renewable Energy Laboratory Turbine Archive (NREL) (NREL, 2020) and Wind Turbine Model database (WTM) (WTM, 2020). The airborne data are available

at PANGEA repository (Baerfuss et al., 2019). ERA5, ERA-Interim reanalysis, coastDat data are available via DKRZ data services at https://docs.dkrz.de/doc/dataservices/index.html. ASCAT data are available at https://www.remss.com/mission/ascat. Wind data from FINO platforms are available at the Bundesamt für Seeschifffahrt und Hydrographie (BSH) (BSH, 2023).

*Author contributions.*   This study was conceptualized by AE and BG with data curation, formal analysis, investigation, methodology, software, validation and visualization carried out by AE Resources were provided by BG and NA The original draft was written by AE, with

subsequent review and editing by BG, NA and CS Project conceptualization and funding acquisition were led by BG

*Competing interests.*   The authors declare no conflicts of interest.

*Acknowledgements.*   The authors would like to acknowledge the German Climate Computing Center (DKRZ) for providing computational resources. ERA5/ERAInterim data reformatted by the CLM community, provided via the DKRZ data pool, were used. We thank the CLM Community for their assistance and collaboration. We thank the Federal Ministry for Economic Affairs and Energy (BMWi) and the Federal

Agency for Shipping and Sea for the FINO data, and the Wind Park Far Field (WIPAFF) project for providing the first in situ airborne atmospheric observational data of the offshore wind farms. C-2015 ASCAT data are produced by Remote Sensing Systems and sponsored by the NASA Ocean Vector Winds Science Team. Data are available at www.remss.com. Thanks to ICDC, CEN, University of Hamburg for data support. The work is funded by the German Federal Ministry of Education and Research in the project H$_2$Mare under the number 03HY302J. NA also acknowledges the support from the German Federal Ministry of Education and Research (BMBF) under project CoastalFutures

(03F0911A), a project of the DAM Research Mission sustainMare – Protection and Sustainable Use of Marine Areas.





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
