# Peer review of "Uncertainty in Offshore Wind Power Forecasts: A Regional Climate Modeling Approach for the North Sea"

_Wind Energy Science, 2025_

## Referee Comment (RC1)

**Review of "Uncertainty in Offshore Wind Power Forecasts: A Regional Climate Modeling Approach for the North Sea" by Alberto Elizalde, Naveed Akhtar, Beate Geyer, and Corinna Schrum, submitted to Wind Energy Science**

The manuscript describes the modeling efforts by the team to quantify the uncertainty in future wind power production as predicted with the COSMO-CLM model over the North Sea, focusing on uncertainties due to the choice of the reanalysis dataset for initial and boundary conditions (6%), the turbine density (2%), and the wind speed at different turbine heights (5%). The uncertainties as defined in the paper are hard to fully understand and somewhat questionable in their interpretations, as discussed in more detail below.

The manuscript is exceptionally well written and logically structured, a pleasure to read really; the scientific approach is solid and the results are relevant, although it is difficult to tell whether they are extendable to other forecasting tools or they are specific to the COSMO-CLM alone. An important innovation of this study is the use of different turbine models and types for the various wind farms, as close as possible to the actual installed type for existing farms or to the most advanced type for the planned ones. As far as I know, this has not been done before and it is a valid approach.

In my opinion, the paper should be published after the issues below are addressed.

**Major issues**

- L. 160: there are intense discussions in the community about the correction factor for C\_TKE. Ignoring it entirely seems untimely. Even though its effects on power production are probably small (based on conference talks I attended), the uncertainty introduced might be of the order of 2-5%, similar to the magnitude of the uncertainties addressed here. I recommend running at least a period per season (with just one reanalysis perhaps) with a reduced value of C\_TKE to have a sense of its impact.
- 2. Table 4 and relevant text: the results of the Chronological case are dramatically different (and better according to observations) than those from the other runs in 2012-2018. I thought that this did not make sense, then I realized that the authors did not conduct a correct validation here and therefore should redo this part. Two wind farms were installed in 2015 near FINO1 and the Chronological run had no wind farms from 2012 to 2015, thus low bias, whereas the other runs incorrectly had those wind farms over those 4 years, thus high bias. Similarly at FINO3, 3 wind farms were installed in 2015 and 2017, thus similar issues. The validation must be conducted only during the last year, 2018, in which no new wind farms were added. My expectation is that the biases will be very similar among the various runs.
- 3. Table 6 and relevant text: unfortunately the analysis and interpretation of these results is not correct. The difference between the two No-wake runs is not the effect of "wind conditions by taller turbines" (150 m vs. 90 m) alone, because two different rotor diameters D were used too (240 m vs. 120 m). To assess the impact of hub height H alone, the 3.6 MW run should have been compared with one identical but with a hub height of 150 m instead of 90 m. Only that way the only power output difference would have been due to the different wind conditions at higher hub height. The difference value

of 5.3% is a combination of hub height and diameter combined, in the absence of wake effects... not sure what that really is, actually. Similarly, the difference between wake losses of -15.5% and -13.5%, thus 2%, is not due to turbine density alone (which would depend on D only), but also on H. All that we can conclude from Table 6 is that the two runs with wakes and different H and D give a higher output by 7.4% for the larger turbine, resulting from a combination of H and D effects that cannot be detangled with the current runs.

**Minor issues**

- 4. L. 75-85: about the issue of the power coefficients. Fig. 1 shows the issue well: the power curve (solid line) gives a different power from that calculated by multiplying the hub-height wind speed by the power coefficient at that speed (dash-dotted line) and the corrected curve is the dashed one. There seems to be only one correction factor for the entire curve from Table 1. Wouldn't it make more sense to correct each value of the power coefficient individually at each wind speed, rather than using one blanket value for all speeds? This is not a big deal and I am not requesting this work, just curiosity and the fact that the weird wiggles and the bumps at high speeds remain and are not too realistic.
- 5. L. 99: are ASCAT data once a day only?
- 6. L. 142: what is the rotated North Pole?
- Table 2 and text: how did you calculate the power without wake losses in the "Control 3.6 MW" and "Control 15 MW" scenarios? I suspect it is just the Control run postprocessed to calculate an (unrealistic) output offline if all turbines were front-row.
- 8. L. 164-167: please explain how the large (15 MW) and small (3.6 MW) turbines were arranged over the same wind farm areas for the 3.6 MW and 15 MW cases. What spacing (8Dx8D? 10Dx10D?) was assumed?
- 9. L. 175 and other occurrences throughout: using two different (but not entirely independent) datasets like ERA5 and ERA Interim does not quantify the uncertainty due to "atmospheric conditions", rather that due to the IC/BC. Basically, any bias in the model chosen for IC/BC is transferred to the simulations, thus this uncertainty reflects the importance of the IC/BC. Please correct the text throughout including the abstract.
- 10. L. 187: show the equation for the bulk Richardson number and explain how you obtained all its terms from the model values and levels.
- 11. Table 4: this table must be redone as mentioned at item 2) above, but if the p-value equals 0.0 for all cases, then it does not need to be shown. However, I recommend using more digits, at least 2 after the decimal separator.
- 12. Figure 5 and relevant text: a definition of what the "anomalies for wind speed" are is necessary. Without it, it appears that the results of the simulation were rather poor and that the wake was exaggerated in lateral extent and length. Please describe what you are showing (anomalies with respect to what?) and why this case was chosen, keeping in mind that it is not convincing of the performance of the wind farm parameterization.

- 13. Figure 7b: this flow does not appear to be from the 183-193 direction. In addition, these results appear very noisy, with huge (unrealistic) accelerations to the north east of the big wind farm cluster. Perhaps numerical noise?
- 14. L. 283 and a few other instances: there is no way an atmospheric flow could be laminar, the Reynolds' numbers are of the order of millions.
- 15. L. 370-371: the wake results were definitely not "well represented", see also comment 12) above.
- 16. L. 383: should 58 GW be 62.6 GW?
- 17. L. 380-383: we already discussed that the actual values of the uncertainties here are not correct and do not really represent what is stated. Nonetheless, there seems to be an assumption here that the uncertainties can be linearly summed up, 5+2+6=13%, which I am not sure about. In fact the power output difference is 74-63=11 GW, which I am not sure about. In fact which is 11/150=7.3%, not 13%. Even if we kept the wrong value of 58 MW, then 74-58=16 GW, not 20 GW as stated on I. 383. All these inconsistencies are due to the wrong interpretations and calculation of the individual uncertainties and to the wrong assumption that they would be linear.